# Photolysis of Fluorinated Graphites with Embedded Acetonitrile Using a White-Beam Synchrotron Radiation

**DOI:** 10.3390/nano12020231

**Published:** 2022-01-11

**Authors:** Galina I. Semushkina, Yuliya V. Fedoseeva, Anna A. Makarova, Dmitry A. Smirnov, Igor P. Asanov, Dmitry V. Pinakov, Galina N. Chekhova, Alexander V. Okotrub, Lyubov G. Bulusheva

**Affiliations:** 1Nikolaev Institute of Inorganic Chemistry SB RAS, 3, Acad. Lavrentiev Ave., 630090 Novosibirsk, Russia; fedoseeva@niic.nsc.ru (Y.V.F.); asan@niic.nsc.ru (I.P.A.); pinakov@niic.nsc.ru (D.V.P.); chekhova@niic.nsc.ru (G.N.C.); spectrum@niic.nsc.ru (A.V.O.); 2Physikalische Chemie, Institut für Chemie und Biochemie, Freie Universität Berlin, 14195 Berlin, Germany; anna.makarova@fu-berlin.de; 3Institut für Festkörper- und Materialphysik, Technische Universität Dresden, 01069 Dresden, Germany; dmitry.smirnov@helmholtz-berlin.de

**Keywords:** fluorinated graphite, acetonitrile, non-monochromatized synchrotron radiation, photolysis, XPS, NEXAFS

## Abstract

Fluorinated graphitic layers with good mechanical and chemical stability, polar C–F bonds, and tunable bandgap are attractive for a variety of applications. In this work, we investigated the photolysis of fluorinated graphites with interlayer embedded acetonitrile, which is the simplest representative of the acetonitrile-containing photosensitizing family. The samples were continuously illuminated in situ with high-brightness non-monochromatized synchrotron radiation. Changes in the compositions of the samples were monitored using X-ray photoelectron spectroscopy and near-edge X-ray absorption fine structure (NEXAFS) spectroscopy. The NEXAFS N K-edge spectra showed that acetonitrile dissociates to form HCN and N_2_ molecules after exposure to the white beam for 2 s, and the latter molecules completely disappear after exposure for 200 s. The original composition of fluorinated matrices CF_0.3_ and CF_0.5_ is changed to CF_0.10_ and GF_0.17_, respectively. The highly fluorinated layers lose fluorine atoms together with carbon neighbors, creating atomic vacancies. The edges of vacancies are terminated with the nitrogen atoms and form pyridinic and pyrrolic units. Our in situ studies show that the photolysis products of acetonitrile depend on the photon irradiation duration and composition of the initial CF_x_ matrix. The obtained results evaluate the radiation damage of the acetonitrile-intercalated fluorinated graphites and the opportunities to synthesize nitrogen-doped graphene materials.

## 1. Introduction

Fluorinated graphites are layered carbon materials possessing a good chemical, mechanical, and thermal stability [1,2]. Fluorination of graphite using inorganic fluorides at room temperature produces compounds with a composition CF_x_, where x is usually below 0.5 [3,4]. The molecules from the reaction media fill the space between the fluorinated layers and they can be replaced by other inorganic or organic guests [5]. Since such compounds are stable at ambient conditions for a long time, the fluorinated graphites are considered as containers for the storage and transport of volatile and hazardous substances [6].

The distance between fluorinated layers varies from ~0.6 to ~1.5 nm depending on the size and packing of the guest molecules [7]. A very weak (if any) interaction between the layers causes a two-dimensional (2D) magnetic behavior of these materials [8]. Their exfoliation in appropriate solvents allows producing thin films of the fluorinated graphene layers [9], which are promising materials for gas sensors and energy applications [9,10]. Guest molecules affect the thermal stability of the fluorinated graphite compounds [5] and the structure of the exfoliated graphene-like materials, particularly, their specific surface area [11] and functional composition [12].

Fluorinated graphites CF_x_ have a bandgap of about 2.5–3.0 eV, when x is between 0.4 and 0.5 [13,14]. They are transparent for visible light, however, the optical properties depend on the guest nature [15]. Due to a large energy gap, fluorinated graphites possess photoluminescence [16,17] and may have a perspective as optical elements, sensors, and for photo-chemotherapy. To clarify the feasibility of these applications, the photostability of the compounds should be studied.

The interlayer space of fluorinated graphite CF_x_ (x ≤ 0.5) is easily occupied by acetonitrile [18,19]. This molecule contains an element (nitrogen) that is absent in CF_x_ and therefore is a convenient probe for studying host-guest interactions under illumination. Photolysis of acetonitrile attracts attention because these molecules are present in interstellar medium [20] and understanding of the acetonitrile fragmentation by ionizing radiation can shed light on the early stages of stars formation [21]. A study of the photolysis of gaseous acetonitrile using synchrotron radiation (SR) in the 7–22 eV range revealed the formation of CN and CH species [22]. An increase of the photon energy to 42 eV allowed for additional registration of the signals from C_2_H_2_^+^, C_2_N^+^ ions and a weak signal from N^2+^ [20]. The ultraviolet (UV) irradiation of an H_2_O:CH_3_CN ice at 20 K yielded a large number of nitriles [23]. Experiments show that the mechanism of photolysis of acetonitrile depends on the energy and power of the exciting radiation as well as its chemical environment. A fluorinated graphite matrix may also affect the photo-induced decomposition of the guest molecules.

The radiation of synchrotron light sources covers a wide range of photon energies, and the photon beam is characterized by high intensity and small focus size. Changes in the composition and electronic state of a compound after exposure to SR can be immediately recorded using X-ray photoelectron spectroscopy (XPS) and near-edge X-ray fine structure (NEXAFS) spectroscopy. The use of tunable SR makes it possible to excite certain electron transitions and study the fragmentation of a compound depending on these states [24]. Non-monochromatized SR light includes a wide range of wavelengths from infrared to hard X-rays with very high intensities. The effect of this polychromatic light on a material determines the material’s resistance to wide-range irradiation, which is important for assessing its possible radiation damage. Illumination with non-monochromatized SR light is also effective for the synthesis of various nanomaterials, for example, gold nanoparticles [25] and the metastable form of carbon, carbyne [26]. It has been shown that this method is useful for tuning the functional composition of carbon nanotubes [27] and graphene materials [28].

This work is aimed at an in situ XPS and NEXAFS study of fluorinated graphites with embedded acetonitrile molecules after continuous illumination with a high-intensity polychromatic photon beam (zero-order light from the dipole beamline of the BESSY II synchrotron radiation facility). We investigated two samples differing in the content of fluorine and acetonitrile. Density functional theory (DFT) calculations of a fluorinated graphene fragment are used to interpret experimental data.

## 2. Materials and Methods

### 2.1. Materials

Purified natural graphite (the Zavalevskoe deposit, Ukraine) was used as a starting material. The typical size of graphite crystallites was 0.4 × 0.3 × 0.02 mm. First, graphite was activated in saturated Br_2_ vapor for 2 days and then the weighed samples were located in Teflon reactors over a liquid mixture of Br_2_ and BrF_3_. The reactors were hermetically closed and kept at room temperature. The synthesis products were washed with Br_2_ to remove residual BrF_3_ and then many times with acetonitrile. The acetonitrile treatment was completed when the washout became colorless, which meant that Br_2_ was removed from the fluorinated graphite interlayer space and replaced with CH_3_CN. Finally, the samples were dried in a N_2_ flow to their constant weights.

The content of carbon, fluorine, bromine, and nitrogen in the samples was determined from the analysis of products of high-temperature destruction of samples in an oxygen atmosphere [19]. According to the obtained data, the composition of the yellow sample (49 days with 13.89 wt% BrF_3_ in Br_2_) can be represented as CF_0.5_Br_0.005_ 0.070CH_3_CN and the green-brown sample (87 days with 5.04 wt% BrF_3_ in Br_2_) as CF_0.3_Br_0.005_ 0.054CH_3_CN. Bromine found in these compounds forms covalent bonds with carbon edges of graphite domains. Below, the studied samples will be denoted CH_3_CN@CF_0.5_ and CH_3_CN@CF_0.3_.

### 2.2. Measurements

XPS spectra of initial samples were measured on a Specs PHOIBOS 150 spectrometer (Specs GmbH, Berlin, Germany) using an Al K_α_ excitation radiation (1486.7 eV). The spot size of the photon beam was about 3 mm. The Casa XPS 2.3.15 software (Casa Software Ltd., Teignmouth, UK) was used for data processing. The C 1s, F 1s, and N 1s spectra were fitted by a product of Gaussian–Lorentzian (7:3) peaks after subtraction of a Shirley background. The binding energies were calibrated to the C(sp^2^) component energy at 284.5 eV.

Irradiations of samples by non-monochromatized SR light (white beam) and subsequent XPS and NEXAFS experiments were carried out at the Russian-German dipole beamline (RGBL Dipole, BESSY II, Berlin, Germany) operated by Helmholtz-Zentrum Berlin für Materialien und Energie. The total light intensity can be estimated as 50 mJ/cm^2^ [27]. The samples are non-conducting; therefore, they were deposited on copper substrates with a scratched surface (roughness ~100 μm) in the thinnest possible layers. The substrates were fixed on a holder and placed in a vacuum chamber providing a residual pressure of 10^−10^ mbar. After acquiring the NEXAFS spectra in total electron yield (TEY) mode, the samples were exposed to a non-monochromatized photon beam for a certain period, and XPS and NEXAFS spectra were recorded directly for the irradiated spots (~1 × 1 mm). The measurements of the spectra accompanied each step of the sample irradiations. The irradiation experiments were repeated at different spots of the samples, and they showed the same trend in the spectral modifications depending on the exposure time. XPS spectra were excited by a photon energy of 830 eV. The binding energies were aligned to the position of the Au 4f_7/2_ line at 84 eV recorded from a clean Au foil. Mass-spectra were registered upon an irradiation of CH_3_CN@CF_0.3_ in a scan mode for *m/z* (~30 scans during 4 s). The residual gas analyzer Extorr XT100M (Extorr Inc., New Kensington, PA, USA) was operated with an electron impact ionizer with an energy of 70 eV.

### 2.3. Calculations

DFT calculations were carried out using the three-parameter hybrid Becke and Lee-Yang-Parr exchange-correlation functional (B3LYP) [29,30] implemented in the program package Jaguar (Jaguar, version 10.3, Schrödinger, Inc., New York, NY, USA, 2019). Atomic orbitals were described by the 6-31G* basis set.

A graphene fragment of a C_96_ composition and D_6h_ symmetry was taken to construct the fluorinated models. Saturation of dangling bond of an edge carbon atom by one fluorine atom yielded the C_96_F_24_ model. An attachment of fluorine atoms to both sides of the basal graphene plane, like in fluorographene [31], and bonding of an edge carbon atom with two fluorine atoms produced the C_96_F_120_ model. The geometries of the models were optimized by an analytic gradient method to default convergence criteria. Then, we removed 34 central fluorine atoms from the C_96_F_120_ fragment to form aromatic and polyene carbon areas in partially fluorinated graphene according to that observed experimentally [32]. The obtained partially fluorinated C_96_F_86_ model was optimized at fixed positions of the boundary atoms.

Theoretical NEXAFS C K- and F K-edge spectra were plotted using the results of DFT calculations of the fluorinated models, where a carbon atom or a fluorine atom was replaced by an atom of nitrogen or neon, respectively. This so-called (Z + 1) approximation accounts for the effect of the core level hole on spectral profile [33]. To compensate for an increase in the number of valence electrons, the system charge was +1. The (Z + 1) approximation was used for the selected carbon or fluorine atoms located in structurally non-equivalent positions. The geometries of the structures with a neon were not optimized to avoid detachment of the neon atom. Spectral intensities were calculated as the sum of the squares of the coefficients, with which the atomic orbitals of nitrogen or neon participate in the formation of unoccupied molecular orbitals (MOs). The calculated intensities were broadened by Lorentz functions with a variable width of 1.4–4.0 eV, increasing with the photon energy, the spectral background was described by an arctan function [34]. X-ray transition energies were determined as the difference between the Kohn-Sham energies of the virtual MOs of the models calculated within the (Z + 1) approximation and energy of the core levels of the selected carbon or fluorine atoms, taken from the calculation of the ground state of the fluorinated model. The spectrum for a central nitrogen atom in the C_96_F_24_ model calculated within the (Z + 1)-approximation (Appendix A) was aligned to the experimental C K-edge spectrum of graphite by the position of π* and σ* peaks. The obtained scaling formula was used to calibrate the energy of other theoretical C K-edge spectra. The calibration of energy for theoretical F K-edge spectra was done from the comparison of the calculated spectrum for the C_96_F_120_ model and the experimental spectrum of fully fluorinated graphite (CF)_n_ (Appendix A).

## 3. Results

### 3.1. XPS C 1s and F 1s Spectra

XPS measurements were used to evaluate the changes in the composition of CH_3_CN@CF_0.3_ and CH_3_CN@CF_0.5_ samples before and after illumination with high-brilliance non-monochromatized SR light. The content of the elements was determined from the survey spectra (not shown) taking into account atomic subshell photoionization cross-sections of elements at a given excitation energy. Atomic concentrations of main elements are collected in Table 1. The content of fluorine and nitrogen in the two studied samples differs by a factor of two in line with the data of elemental analysis. The XPS-derived F/C ratio is 0.16 for CH_3_CN@CF_0.3_ and 0.37 for CH_3_CN@CF_0.5_. Since XPS is a surface-sensitive method, it detects the low content of fluorine in the upper surface layers of the samples as a result of their partial de-fluorination due to the contact with H_2_O present in laboratory atmosphere [35]. Higher oxygen content on the surface of CH_3_CN@CF_0.3_ than for CH_3_CN@CF_0.5_ may indicate an easier replacement of fluorine by oxygen in this sample. The weakness of C–F bonds in CH_3_CN@CF_0.3_ results in almost complete removal of surface fluorine under the photon irradiation. While the CH_3_CN@CF_0.5_ sample keeps about 3 at% of fluorine even after 200-s exposure to polychromatic synchrotron light. XPS also detects nitrogen from CH_3_CN molecules in both initial samples and after each step of the irradiation.

Figure 1 compares XPS C 1s and F 1s spectra of the samples. C 1s spectra of initial CH_3_CN@CF_0.5_ and CH_3_CN@CF_0.3_ are fitted by four components (Figure 1a,b). The binding energies and the relative areas of the components are listed in Appendix A. A weak component at ~287 eV corresponds to carbon in guest acetonitrile molecules [36]. The low-energy component at 284.5 eV originates from sp^2^–carbon areas remaining in the fluorinated layers and its intensity is higher for the CF_0.3_ matrix than for CF_0.5_. The peaks at 288.7 and 286.1 eV in the CH_3_CN@CF_0.5_ spectrum characterize carbon atoms covalently bonded to fluorine (C–F) and located at CF groups (C–CF), respectively [37]. These peaks are downshifted by 0.7 eV for CH_3_CN@CF_0.3_ due to the weakening of C–F bonds [38,39]. The intensity of the C–F peak correlates with fluorine content in the samples (Table 1). The ratio of the C–F component to the total area of the C 1s spectrum gives matrix stoichiometry CF_0.24_ for the CH_3_CN@CF_0.3_ sample and CF_0.43_ for the CH_3_CN@CF_0.5_ sample. The C–CF/C–F ratio gives the average number of bare carbon atoms near CF groups. The ratio 1 for CH_3_CN@CF_0.5_ (Table 1) indicates an average of one bare carbon neighbor for a CF group. Such a ratio can be realized when CF chains alternate with bare carbon chains [40,41]. An increase in the ratio value for CH_3_CN@CF_0.3_ is associated with the shortening of CF chains and increase in numbers of two bare carbon neighbors for CF groups located at the edges of short CF chains.

The relative intensity of the C–F component decreases in the C 1s spectra of irradiated samples (Figure 1a,b) due to the removal of fluorine. The shift of the C–F and C–CF components to lower binding energies indicates the weakening of C–F bonds as compared to those in the initial samples. New components located at ~289.5 and ~292 eV are especially noticeable in the spectrum of CH_3_CN@CF_0.5_ after 200 s of the irradiation. They are assigned to carbon bonded with two (CF_2_) and three fluorine atoms (CF_3_) [42]. Thus, white beam partially destroys graphitic lattice. The detached carbon and fluorine atoms are combined with the CF_2_ and CF_3_ groups that bind to the edges of vacancies [43]. Analysis of the XPS C 1s spectra reveals that the composition of the CH_3_CN@CF_0.5_ and CH_3_CN@CF_0.3_ samples irradiated for 200 s is CF_0.18_ and CF_0.10_, respectively.

XPS F 1s spectra of initial CH_3_CN@CF_0.5_ and CH_3_CN@CF_0.3_ exhibit a single peak at 687.2 and 686.6 eV, respectively (Figure 1c,d). These binding energies correspond to fluorine covalently bonded with carbon [15,44]. The exposure of the samples to the non-monochromatized light leads to the emergence of fluorine states possessing higher binding energies. The F 1s components at 689.0 and 691.0 eV can be attributed to fluorine in CF_2_ and CF_3_ groups [45], or the atoms located in densely fluorinated regions [44] like in (CF)_n_. However, the C–CF/C–F ratio in the XPS C 1s spectra of the samples (Table 1) indicates that most CF groups have one or two bare carbon atoms as their neighbors, and this differs from the fluorine arrangement in (CF)_n_. Amounts of CF_2_ and CF_3_ groups are larger in the irradiated CH_3_CN@CF_0.5_ than in the irradiated CH_3_CN@CF_0.3._

### 3.2. NEXAFS C K-Edge and F K-Edge Spectra

NEXAFS spectra measured before and after sequential irradiation of CH_3_CN@CF_0.5_ and CH_3_CN@CF_0.3_ for 20, 80, and 200 s are presented in Figure 2. The difference in the binding energies of the XPS F 1s peak and the C–F component of the XPS C 1s spectrum (Figure 1) is used for the energy alignment of NEXAFS C K- and F K-edge spectra of the particular sample.

NEXAFS C K-edge spectra of all samples exhibit π* and σ* resonances at 285.1 and 291.9 eV (Figure 2a,b) assigned to the electron transitions from C 1s levels onto unoccupied π-type and σ-type states for sp^2^-hybridized carbon, respectively [46,47,48]. The peaks, which appeared between these resonances at 287.8 and 288.8 eV and labeled C_1_ and C_2_, correspond to carbon bonded with fluorine [49,50]. In the spectrum of starting CH_3_CN@CF_0.5_, these peaks are more prominent, while the π* resonance has the lowest intensity (Figure 2a). The letter C denotes the position of σ*-edge for the fluorinated areas because it coincides with the last intense peak (labeled F) of the F K-edge spectra (Figure 2c,d). The shoulder F_1_ at 686.5 eV and the peak F_2_ at 687.4 eV align with peaks C_1_ and C_2_ of the C K-edge spectra and therefore they refer to the C–F bonds. The illumination of CH_3_CN@CF_0.5_ and CH_3_CN@CF_0.3_ samples with polychromatic synchrotron beam results in the suppression of C_1_ and C_2_ peaks in C K-edge spectra and F_1_ and F_2_ peaks in F K-edge spectra and the growth of relative intensity of π* resonance from sp^2^-carbon. The changes are stronger with increasing exposure time and correlate with the behavior observed in the XPS C 1s spectra of the samples (Figure 1).

To interpret NEXAFS C K-edge and F K-edge spectra in detail, NEXAFS spectra for structurally nonequivalent carbon and fluorine atoms present in the partially fluorinated graphitic monolayer are constructed (Figure 3). The spectra of the starting and irradiated for 200 s CH_3_CN@CF_0.5_ sample are chosen for the modeling (Figure 3a,b). The calculated fluorinated graphene fragment is shown in Figure 3c. Theoretical spectra are constructed for the carbon and fluorine atoms from CF groups surrounded by three (CF-3), two (CF-2), one (CF-1), and none (CF-0) fluorinated carbon atoms. We also calculate the C K-edge spectra for bare carbon atoms from polyene-like chain (C-ch) and aromatic naphthalene-like area (C-ar).

Comparison of the C K-edge spectrum of initial CH_3_CN@CF_0.5_ with the calculated spectra shows that energy of π* resonance corresponds to the position of the low-energy intense peak in the spectrum of C-ch (Figure 3a). This result indicates that most of the sp^2^-hybridized carbon atoms in the fluorinated CF_0.5_ layers form polyene-like chains, that is in agreement with the previous data [40,51]. Peak C_1_ originates from carbon in isolated CF groups (CF-0), while carbon atoms from CF-1 and CF-2 groups, where CF groups have one and two CF neighbors, respectively, contribute to the peak C_2_ in the experimental spectrum. Fluorine atoms from CF-1 and CF-2 groups are responsible for intense peaks F_2_ and F_3_ in the F K-edge spectrum of CH_3_CN@CF_0.5_ (Figure 3b). Shoulder F_1_ is assigned to fluorine from isolated CF-0 groups. Analysis of MOs calculated in the (Z + 1)-approximation reveals that spectral features F_1_, F_2_, and F_3_ correspond to C–F bonds of σ*-type (Appendix A). The difference in energy is due to the different local distribution of electron density between this bond and the neighbors. The high-energy peak F is formed by an overlapping of F 2p_x,y_ orbitals with neighboring C–C σ-bonds.

The cumulative theoretical C K-edge spectrum obtained by summing the spectral intensity of carbon from C-ch, CF-0, CF-1, and CF-2 taken in a ratio of 1.8:1:1:1.8 perfectly repeats the shape of the experimental spectrum of initial CH_3_CN@CF_0.5_ (two upper curves in Figure 3a). The cumulative F K-edge spectrum being a sum of the theoretical spectra of fluorine from CF-0, CF-1, and CF-2 taken in a proportion of 1:1:1.8 also agrees well with the experimental spectrum of CH_3_CN@CF_0.5_ (two upper curves in Figure 3b). CF-3 groups with three CF neighbors are not necessary to define all spectral features; probably, they are hardly formed in the synthesis conditions used. Fluorine distribution in the layers of a CF_0.5_ composition is mainly realized as CF chains separated by polyene-like carbon chains.

Exposure of CH_3_CN@CF_0.5_ to white beam for 200 s causes an increase and broadening of π* resonance and a significant decrease in the intensity of C_1_ and C_2_ peaks of the C K-edge spectrum (Figure 3a). To describe this spectral profile, the spectra of CF-0, C-ch, C-ar, and central atom in the graphene model (Appendix A) are taken in a ratio of 1:1:2:2. The F K-edge spectrum of the irradiated CH_3_CN@CF_0.5_ shows mainly a decrease in the intensity of F_2_ peak (Figure 3b) and only isolated fluorine atoms from CF-0 groups are needed to simulate the experimental profile. These results indicate that long-term irradiation of the fluorinated graphitic layers leads to their strong defluorination. The remaining fluorine atoms are separated from each other. A significant removal of fluorine occurs after the first 20-s irradiation and it is more pronounced for the CH_3_CN@CF_0.3_ sample (Figure 2).

### 3.3. Electronic State of Nitrogen

Electronic state of nitrogen from acetonitrile molecules embedded between the fluorinated graphitic layers is revealed using XPS N 1s and NEXAFS N K-edge spectra. The XPS N 1s spectrum of initial CH_3_CN@CF_0.5_ exhibits a single symmetrical peak at ~399 eV (Figure 4a). The white beam illumination of the sample for 80 s causes the appearance of two new components located at ~398.1 and ~400.5 eV and assigned to pyridinic N and –NH– species in carbon rings (pyrrolic N), respectively [52]. The fraction of the pyrrolic N increases with the irradiation duration. This result indicates that CH_3_CN molecules are decomposed under photon-beam treatment and the released nitrogen and hydrogen atoms are incorporated into the surrounding CF_x_ layers. Insertion of nitrogen into fluorinated graphitic layers was early observed for similar CH_3_CN@CF_x_ samples heated at 250 °C in a vacuum [53].

NEXAFS measurements were performed to examine the initial stages of sample irradiation in more details, thus Figure 4b compares N K-edge spectra of starting CH_3_CN@CF_0.5_ and that after exposure to white beam during 2, 5, 20, 80, and 200 s. The irradiation of CH_3_CN@CF_0.5_ for 2 s already results in degradation of acetonitrile. The pre-edge peak C≡N located in the initial spectrum at ~399.9 eV shifts by 0.3 eV to the low-energy region, its intensity decreases, and new peak around 401.0 eV appears. Our DFT calculations show the shift of the C≡N peak can be attributed to the formation of HCN (Appendix A). The peak at about 401.0 eV corresponds to pyrrolic N species [54] and N_2_ molecules [55]. The intensity of this peak strongly reduces in the N K-edge spectrum of CH_3_CN@CF_0.5_ (Figure 4b) and CH_3_CN@CF_0.3_ (Appendix A) irradiated for 20 s.

NEXAFS N K-edge spectra of starting and irradiated CH_3_CN@CF_0.3_ sample measured in a range of 397.5–403.0 eV are shown with a purpose to study the pre-edge peaks in detail (Figure 4c). The resonance emerging around 401.0 eV is resolved into five peaks characteristic of vibrations of N_2_ molecules [56]. This proves the formation of N_2_ molecules as a result of the photolysis of CH_3_CN and the retention of these molecules between the fluorinated graphitic layers. The content of the trapped N_2_ molecules decreases with continuing irradiation and the molecules are not detected after 80-s irradiation. The peak at ~ 400.6 eV (Figure 4b,c) corresponding to pyrrolic N at the boundaries of vacancies in CF_x_ layers is identified according to the DFT calculations (Appendix A). Incorporation of pyridinic N occurs at the first stages of samples irradiation and raises the shoulder at 398.8 eV in the N K-edge spectra (Figure 4b,c).

Mass spectrum of ion species measured upon the irradiation of CH_3_CN@CF_0.3_ sample is presented in Figure 5. The background ion peaks from residual air and molecular ion peaks are highlighted in black and red, respectively. The signal of CO_2_^+^ ions (*m/z* = 44) arising from the sample surface is taken as ~100%. Note that the amplitude of background H^+^ and H_2_O^+^ ions is an order of magnitude larger than this signal. The spectrum detects the ions being the decomposition products of CH_3_CN molecules. They are CH_3_^+^/NH^+^ (*m/z* = 15), C_2_H_2_^+^/CN^+^ (*m/z* = 26), C_2_H_3_^+^/CHN^+^ (*m/z* = 27), N_2_^+^ (*m/z* = 28), CH_2_N^+^ (*m/z* = 28), CH_2_CN^+^ (*m/z* = 40), CH_3_CN^+^ (*m/z* = 41), and C_2_N_2_^+^ (*m/z* = 52). The ions FCNH^+^ (*m/z* = 47) CF^+^ (*m/z* = 50), and CF_3_^+^ (*m/z* = 69) contain the atoms from fluorinated graphitic layers. A combination of fluorine with carbon may indicate that fluorine is removed along with the lattice carbon.

## 4. Discussion

Fluorinated graphites of the composition CF_0.5_ and CF_0.3_ are insulators, and their XPS spectra are measured in a laboratory spectrometer where charging of sample under X-ray photon exposure is compensated. Analysis of XPS data indicates that C–F bonds are covalent and they are weaker in CF_0.3_ layers. These layers also contain larger fractions of aromatic areas and bare carbon atoms located nearby CF groups as compared to CF_0.5_ layers. The DFT modeling of NEXAFS C K- and F K-edge spectra of initial CF_0.5_ reveals that fluorine atoms in the layers preferably form CF chains, alternating with polyene-like carbon chains. The CF chains are shorter in CF_0.3_ layers.

NEXAFS C K- and F K-edge spectra of both studied samples exhibit a large decrease in the intensity of the peaks corresponding to the C–F bonds already after polychromatic photon beam exposure for 20 s (Figure 2). A probing depth of the spectra acquired in the TEY mode is a few nanometers [38], thus, we estimate that at least ten upper layers lose fluorine. The areas of C–F components in XPS C 1s spectra of both samples decrease by about 2.4 times after irradiation for 200 s (Appendix A). The shape of the F K-edge spectrum of the irradiated CF_0.5_ well corresponds to the electronic state of fluorine in isolated CF groups, i.e., not adjacent to other CF groups (Figure 3b).

A part of fluorine atoms removed from basal graphitic planes is attached to their edges as CF_2_ and CF_3_ groups and amounts of these groups are markedly larger for CF_0.5_ layers (Figure 1a,b). The formation of these groups means that a fluorine atom may leave a fluorinated graphitic layer together with the carbon neighbor. The CF^+^ and CF_3_^+^ ion signals detected in the mass spectrum upon the irradiation of CH_3_CN@CF_0.3_ sample (Figure 5) confirm this. According to our DFT calculations, fluorine atoms located in the center of CF chains have larger binding energies than fluorine atoms at the chain edges (Appendix A). Therefore, the C–F bonds in long CF chains that are present in CF_0.5_ layers are stronger than bonds between bare carbon atoms and fluorinated ones (C−CF). The latter C−CF bonds break more easily, producing vacancies in the carbon network. The edge C–F bonds, which are predominant in short CF chains of CF_0.3_ layers, dissociate under the radiation and this explains the less efficient destruction of C–C bonds observed for this sample.

NMR study of fluorinated graphites with embedded acetonitrile molecules revealed that interactions between the guests and matrices have van-der-Waals character [57]. Such weak interactions should not influence the radiation stability of the constituents. According to NEXAFS N K-edge spectra the CH_3_CN molecules located between fluorinated graphene layers dissociate under irradiation for 2 s (Figure 4b,c). The features detected in the spectra measured at various stages of the irradiation are assigned to HCN and N_2_ molecules, and pyridinic and pyrrolic nitrogen atoms at vacancies edges of CF_x_ layers. The fractions of these nitrogen forms are determined from the decomposition of N K-edge spectra (Appendix A).

The evolution of nitrogen forms in CH_3_CN@CF_0.5_ and CH_3_CN@CF_0.3_ with the exposure time is illustrated in Figure 6a,b. The main product of the CH_3_CN photolysis is HCN molecules. Their preferable formation according to the path CH_3_CN→ CH + H + HCN was previously observed under the action of UV light [23], electron beam [20], and photon beam [22]. The HCN concentration after the first stage of the irradiation for 2 s is ~64% for CH_3_CN@CF_0.5_ and ~57% for CH_3_CN@CF_0.3_, and it reduces to ~49% for CH_3_CN@CF_0.5_ and ~47% for CH_3_CN@CF_0.3_ after the 200-s exposure. Since the pyrrolic N is detected in the NEXAFS N K-edge spectra measured for the samples after the 80-s exposure, we suppose that decomposition of HCN with prolonged irradiation of samples contributes to the formation of this kind of nitrogen species.

The initial photolysis of CH_3_CN molecules yields also N_2_ molecules and pyridinic N atoms (Figure 6a,b). A weak signal of nitrogen ions was observed when gaseous CH_3_CN was photoionized by the monochromatic SR beam with a photon energy of 42 eV [20]. The CF_x_ matrix probably facilitates the abstraction of the nitrogen atoms from acetonitrile molecules in our case. Mass spectrum of photo-induced ions detects the N_2_^+^ signal (Figure 5). The N_2_ molecule fraction is twice the fraction of pyridinic N developed in CF_x_ layers. However, with an increase in the exposure time, N_2_ content decreases, which is accompanied by a growth of the content of pyridinic N. This behavior indicates photodissociation of N_2_ and incorporation of the produced nitrogen atoms in the surrounding defective CF_x_ layers. The process is faster for the CH_3_CN@CF_0.5_ sample. The reason may be a larger amount of atomic vacancies in the layers as the XPS C 1s and F 1s spectra of the sample detect (Figure 1).

Figure 6c schematically presents changes that occurred for fluorinated graphene layers with acetonitrile guests under the influence of non-monochromatized SR light. The matrix layers lose a part of fluorine and acquire nitrogen atoms. These atoms are located at the boundaries of vacancies, produced when fluorine atoms are removed from the layer together with carbon. The photolysis of acetonitrile produces N_2_ and HCN molecules. DFT calculations show that these molecules are readily adsorbed on the nitrogen-doped CF_x_ layer. Pyridinic N atoms and CF groups create the preferred positions for HCN and N_2_ molecules, respectively.

## 5. Conclusions

Fluorinated graphites with the composition of the layers CF_0.3_ and CF_0.5_ were synthesized using a fluorinating agent BrF_3_ at room temperature. DFT modeling of NEXAFS C K-edge and F K-edge spectra showed that fluorine atoms form the fluorinated carbon chains alternating with polyene-like carbon chains in CF_0.5_ layers. These chains were shorter in the CF_0.3_ layers, where the CF groups have two bare carbon neighbors on average. The interlayer space of the fluorinated graphites was filled by CH_3_CN. Photolysis of CH_3_CN@CF_0.3_ and CH_3_CN@CF_0.5_ was carried out using the zero-order light from the Russian-German dipole Beamline of the synchrotron source BESSY II. The photon irradiation led to a partial defluorination of the layers and the formation of vacancy defects. The XPS C 1s and F 1s spectra showed that the amount of vacancies is larger in the layers with an initial composition of CF_0.5_. C–F bonds in these layers were stronger than the bonds between the fluorinated carbon and bare carbon neighbor that caused the preferred breakage of the latter bonds. CH_3_CN molecules were completely decomposed during the first two seconds of the SR zero-order light exposure. The main products were HCN and N_2_ molecules and pyridinic N atoms, introduced into the CF_x_ layers at the vacancy boundaries. Upon further irradiation, N_2_ molecules dissociated and the released nitrogen atoms gave mainly pyridinic N defects in the fluorinated graphene layers. This dissociation was faster in the CF_0.5_ layers. The products of HCN photolysis contributed to the formation of pyrrolic N species. The study shows that the products of the photolysis of CH_3_CN depend on the time of irradiation and the fluorine loading of the fluorographitic matrix. Our results can be crucial when using CH_3_CN@CF_x_ systems in environments with intense light from UV to soft X-rays.

## Figures and Tables

**Figure 1 nanomaterials-12-00231-f001:**
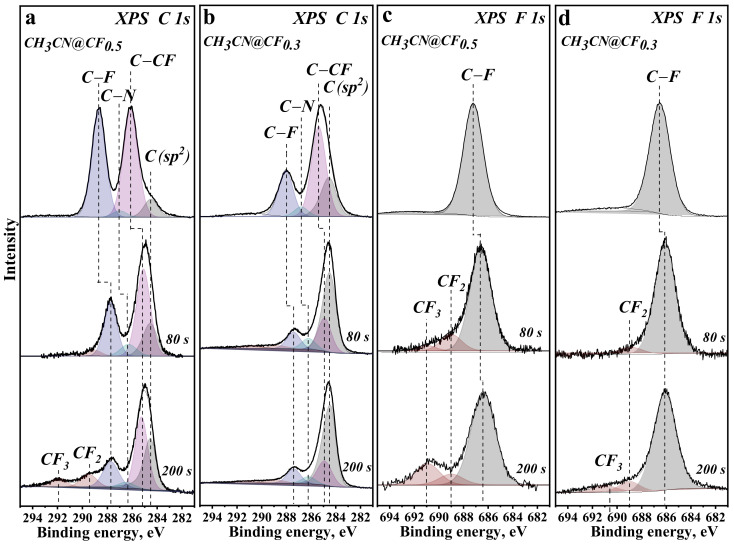
XPS C 1s spectra (**a**,**b**) and F 1s spectra (**c**,**d**) of CH_3_CN@CF_0.5_ (**a**,**c**) and CH_3_CN@CF_0.3_ (**b**,**d**) before and after white beam irradiation for 80 s and 200 s.

**Figure 2 nanomaterials-12-00231-f002:**
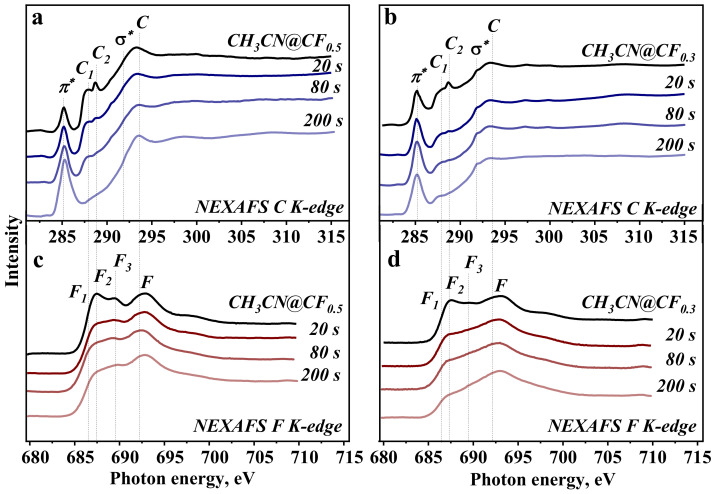
NEXAFS C K-edge (**a**,**b**) and F K-edge (**c**,**d**) spectra of CH_3_CN@CF_0.5_ (**a**,**c**) and CH_3_CN@CF_0.3_ (**b**,**d**) before (upper curves) and after irradiation by white beam for 20 s, 80 s, and 200 s.

**Figure 3 nanomaterials-12-00231-f003:**
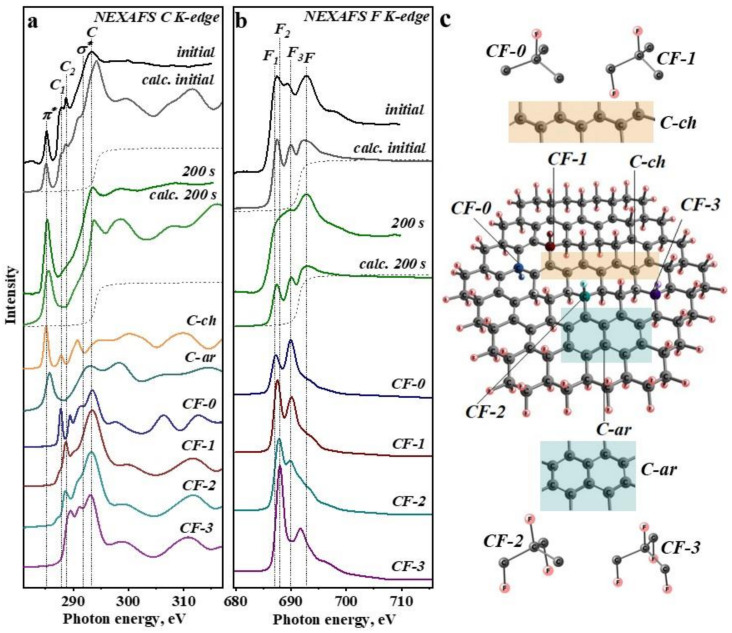
Experimental NEXAFS C K-edge (**a**) and F K-edge (**b**) spectra of CH_3_CN@CF_0.5_ before (black) and after white beam irradiation for 200 s (olive) in comparison with the theoretical spectra calculated for carbon (**a**) and fluorine (**b**) atoms from CF-0, CF-1, CF-2, and CF-3 groups and bare carbon atoms from polyene chain (C-ch) and aromatic area (C-ar) of the partially fluorinated graphene model (**c**).

**Figure 4 nanomaterials-12-00231-f004:**
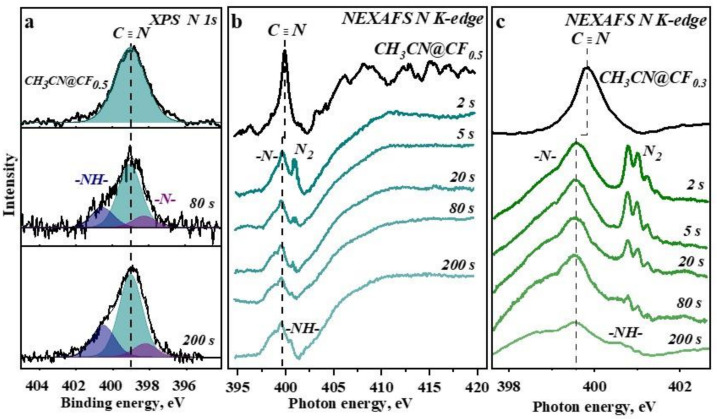
XPS N 1s spectra of initial CH_3_CN@CF_0.5_ and that irradiated for 80 s and 200 s (**a**). NEXAFS N K-spectra of CH_3_CN@CF_0.5_ (**b**) and CH_3_CN@CF_0.3_ (**c**) before and after white beam irradiation for 2 s, 5 s, 20 s, 80 s, and 200 s.

**Figure 5 nanomaterials-12-00231-f005:**
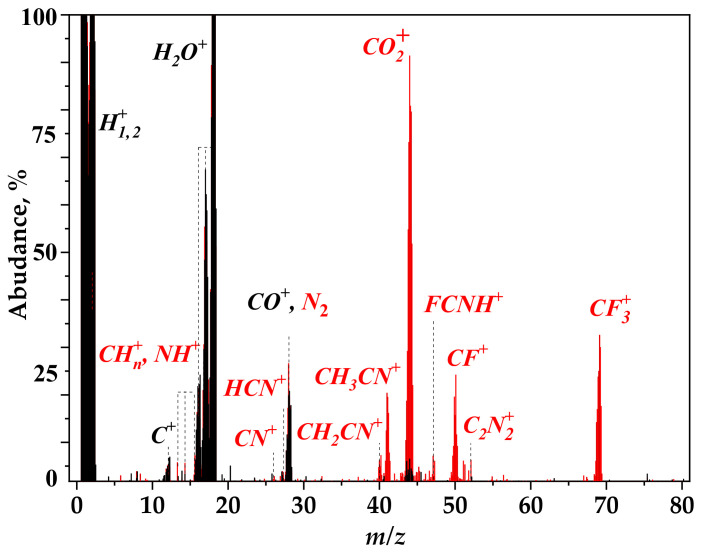
Mass spectrum registered upon white beam irradiation of CH_3_CN@CF_0.3_ sample.

**Figure 6 nanomaterials-12-00231-f006:**
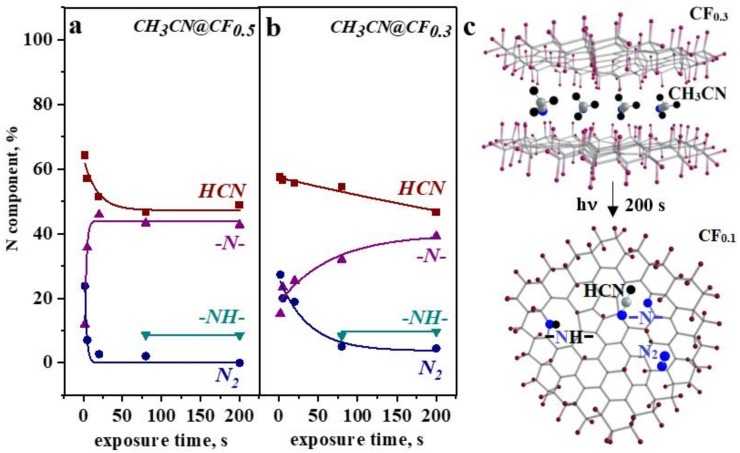
Dependence of the NEXAFS N K-edge components from HCN, N_2_, pyridinic N, and pyrrolic N on the time of exposure of CH_3_CN@CF_0.5_ (**a**) and CH_3_CN@CF_0.3_ (**b**) to white beam light. The model of evolution of CH_3_CN@CF_0.3_ structure and composition under the irradiation (**c**).

**Table 1 nanomaterials-12-00231-t001:** XPS determined content (at%) of main elements in CH_3_CN@CF_0.3_ and CH_3_CN@CF_0.5_ samples before and after exposure to polychromatic synchrotron light for 80 and 200 s. The ratio of the areas of the C–CF to C–F components in the XPS C 1s spectra (last column).

Exposure Time, s	C	F	N	O	C–CF/C–F
CH_3_CN@CF_0.3_					
0	83	13	1	3	1.9
80	98	<1	<1	<1	1.8
200	98	<1	<1	<1	1.9
CH_3_CN@CF_0.5_					
0	71	26	2	1	1.0
80	90	6	3	1	2.0
200	95	3	2	<1	2.2

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
