# Peer review of "Photolysis of Fluorinated Graphites with Embedded Acetonitrile Using a White-Beam Synchrotron Radiation"

_nanomaterials, 2022, doi:10.3390/nano12020231_

Round 1
Reviewer 1 Report
The authors described the Photolysis of fluorinated graphites with embedded acetonitrile using a white-beam synchrotron radiation. The subject of the paper is of great interest for reserachers working with carbon materials and synchrotron radiation. The paper is well-written, however to increase its impact the authors should extend the introduction comparing other reports in the literature. More applications should be mentioned and the advantages of using the white beam synchrotron radiation should be better discussed.
The authors must better explain the fitting of the XPS spectra, more details should be added to the supplementary material.
Author Response
The authors described the Photolysis of fluorinated graphites with embedded acetonitrile using a white-beam synchrotron radiation. The subject of the paper is of great interest for reserachers working with carbon materials and synchrotron radiation. The paper is well-written, however to increase its impact the authors should extend the introduction comparing other reports in the literature. More applications should be mentioned and the advantages of using the white beam synchrotron radiation should be better discussed.
Reply: We thank the Reviewer for the high evaluation of our work and suggestions on how to increase the impact of the paper. The use of synchrotron radiation (SR) light with a certain energy makes it possible to excite particular processes in matter. We conducted irradiation experiments at the soft X-ray Russian-German dipole beamline of BESSY II. Its non-monochromatized beam ranges from infrared light to soft X-rays. Exposing a sample to such polychromatic photon beam of high intensity makes it possible to observe changes in the composition and electronic state of a compound, which can occur in various possible environments, including environmental conditions on the Earth. Such experiments can be useful in assessing the stability of a material used in outdoor and wearable devices. Studying UV resistance is important for optical elements, and data on the radiation damage can be useful in medical applications. Thus, experiments with non-monochromatized SR beam determine the material's resistance to wide-range irradiation. Moreover, the non-monochromatized SR beam is used to synthesize nanomaterials, for example, gold nanoparticles and carbyne. We added the corresponding description in the Introduction: “The radiation of synchrotron light sources covers a wide range of photon energies, and the photon beam is characterized by high intensity and small focus size. Changes in the composition and electronic state of a compound after exposure to SR can be immediately recorded using X-ray photoelectron spectroscopy (XPS) and near-edge X-ray fine structure (NEXAFS) spectroscopy. The use of tunable SR makes it possible to excite certain electron transitions and study the fragmentation of a compound depending on these states [24]. Non-monochromatized SR light includes a wide range of wavelengths from infrared to hard X-rays with very high intensities. The effect of this polychromatic light on a material determines the material's resistance to wide-range irradiation, which is important for assessing its possible radiation damage. Illumination with non-monochromatized SR light is also effective for the synthesis of various nanomaterials, for example, gold nanoparticles [25] and the metastable form of carbon, carbyne [26]. It has been shown that this method is useful for tuning the functional composition of carbon nanotubes [27] and graphene materials [28].”
The authors must better explain the fitting of the XPS spectra, more details should be added to the supplementary material.
Reply: According to the Reviewer request we added the values of binding energies and relative areas of the components obtained from the fitting of C 1s spectra in Table S1. Corresponding modification was indicated in the manuscript text as: “The binding energies and the relative areas of the components are listed in Table S1.”
Details of the XPS spectra analysis were added in the part 2.2. Measurements: “The C 1s, F 1s, and N 1s spectra were fitted by a product of Gaussian-Lorentzian (7:3) peaks after subtraction of a Shirley background.”
Reviewer 2 Report
The paper by Semushkina and coworkers describes a detailed study of photolysis of fluorinated graphite with acetonitrile embedded between the layers using a combination of XPS, NEXAFS and modelling. The work is carefully executed and well-described. The following questions and suggestions require consideration prior to publication.
- Although this is an interesting fundamental study it does not appear to be very relevant to biotechnology applications, as noted in the last sentence of the Conclusions. References 10, 11, 12 for drug delivery, antibacterial treatment and phototherapy applications of fluorinated graphites in the Introduction are all for graphene and in most cases the graphene has been functionalized for biocompatability. Further, the types of drugs or sensitizers used are relatively large molecules compared to acetonitrile. The acetonitrile containing sensitizers that are cited as inhibitors of cancer cells (ref 16-18) are all metal complexes. The authors should provide a more convincing rationale for how this study of a hydrophobic graphite with embedded acetonitrile is relevant to biotechnology applications.
- Section 2.1. The use of “discolored” in the sentence on washing the samples with bromine (lines 72-4) is unclear. Does this mean that the samples were washed until colorless or until the color due to bromine was removed? Discolored would be interpreted as meaning a change from the normal color, which is ambiguous.
How thick are the samples used for the experiments? - Section 2.2. Were XPS measurements taken for different areas of the samples to check for reproducibility? Details on the software used for data analysis and fitting of peaks to multiple components should be provided.
Were the irradiation experiment repeated using multiple sample areas? - The description of the XPS results in Figure 1 is qualitative. Since the peaks have been fit to multiple components, can the fraction of each component be provided, perhaps in the SI?
- Figure 6. The quality of panels a and b is quite poor, particularly compared to the other figures. Please improve.
Typos, English
Line 33, can be replaced by
Line 60, This work is aimed at an in situ…
Line 62, samples differing by…
Line 214, assigned to the electron…
Line 274, the remaining fluorine atoms
Line 286, the released nitrogen..
Line 305, the molecules are not detected after …
Line 416, caused the preferred breakage of the latter bonds…
Author Response
The paper by Semushkina and coworkers describes a detailed study of photolysis of fluorinated graphite with acetonitrile embedded between the layers using a combination of XPS, NEXAFS and modelling. The work is carefully executed and well-described. The following questions and suggestions require consideration prior to publication.
Reply: We are grateful to the Reviewer for helpful comments and will provide detailed answers below.
- Although this is an interesting fundamental study it does not appear to be very relevant to biotechnology applications, as noted in the last sentence of the Conclusions. References 10, 11, 12 for drug delivery, antibacterial treatment and phototherapy applications of fluorinated graphites in the Introduction are all for graphene and in most cases the graphene has been functionalized for biocompatability. Further, the types of drugs or sensitizers used are relatively large molecules compared to acetonitrile. The acetonitrile containing sensitizers that are cited as inhibitors of cancer cells (ref 16-18) are all metal complexes. The authors should provide a more convincing rationale for how this study of a hydrophobic graphite with embedded acetonitrile is relevant to biotechnology applications.
Reply: We agree with the Reviewer, that hydrophilic materials are essential for biotechnology and therefore the oxygen-containing groups are often attached to fluorinated graphitic layers. In this work, we studied the hydrophobic fluorinated graphite samples. The Reviewer is also right that at present our research is of fundamental interest, but, the studied materials possess a wide bandgap and can be used as optical elements. In this regard, their photostability should be determined. Moreover, photodestruction of fluorinated graphites with embedded guests can be used as a method for the synthesis of new nanomaterials. We clarified all these points in the Introduction: “Their exfoliation in appropriate solvents allows producing thin films of the fluorinated graphene layers [9], which are promising materials for gas sensors and energy applications [9,10]. Guest molecules affect the thermal stability of the fluorinated graphite compounds [5] and the structure of the exfoliated graphene-like materials, particularly, their specific surface area [11] and functional composition [12].
Fluorinated graphites CFx have a bandgap of about 2.5 – 3.0 eV, when x is between 0.4 and 0.5 [13, 14]. They are transparent for visible light, however, the optical properties depend on the guest nature [15]. Due to a large energy gap, fluorinated graphites possess photoluminescence [16, 17] and may have a perspective as optical elements, sensors, and for photo-chemotherapy. To clarify the feasibility of these applications, the photostability of the compounds should be studied.”
We chose acetonitrile as a guest in a fluorinated graphite host because this molecule contains nitrogen that is not present in the CFx layers, and this element can be a marker for checking changes in the layers under the action of synchrotron light. The photolysis of acetonitrile was also investigated early, but not in the CFx host. We added a couple of sentences in the Introduction to clarify this issue: “This molecule contains an element (nitrogen) that is absent in CFx and therefore is a convenient probe for studying host-guest interactions under illumination. Photolysis of acetonitrile attracts attention because these molecules are present in interstellar medium [20] and understanding of the acetonitrile fragmentation by ionizing radiation can shed light on the early stages of stars formation [21].”
We also modified the last sentence in the Conclusions: “Our results can be crucial when using CH3CN@CFx systems in environments with intense light from UV to soft X-rays.”
- Section 2.1. The use of “discolored” in the sentence on washing the samples with bromine (lines 72-4) is unclear. Does this mean that the samples were washed until colorless or until the color due to bromine was removed? Discolored would be interpreted as meaning a change from the normal color, which is ambiguous.
How thick are the samples used for the experiments?
Reply: Bromine is a brown liquid while acetonitrile is colorless. The washing of the bromine-intercalated fluorinated graphite replaces Br2 by CH3CN. The initial portion of washout is colored due to bromine and we stop the treatment when the washout is colorless. We corrected the description of the synthesis process as: “The acetonitrile treatment was completed when the washout became colorless, which meant that Br2 was removed from the fluorinated graphite interlayer space and replaced with CH3CN.”
The investigated samples of fluorinated graphite are not electrically conductive. To record XPS and NEXAFS spectra at the synchrotron station, we deposited very thin layers of samples on copper substrates. To increase the adhesion of the sample to the substrate, the surface of the substrate was scratched. As a result, the surface roughness was about 100 μm, and this value is the lower limit of the sample thickness. We added the next sentence to the part 2.2. Measurements: “The samples are non-conducting; therefore, they were deposited on copper substrates with a scratched surface (roughness ~ 100 μm) in the thinnest possible layers.”
- Section 2.2. Were XPS measurements taken for different areas of the samples to check for reproducibility? Details on the software used for data analysis and fitting of peaks to multiple components should be provided.
Were the irradiation experiment repeated using multiple sample areas?
Reply: For the synthesis, a fraction of natural graphite with a size of 0.4 × 0.3 × 0.02 mm was used. Fluorination was carried out at room temperature for a long time (49 days or 87 days for the samples under study), which allows the crystallites to be fluorinated uniformly. The spot size of the photon beam in XPS measurements on a laboratory spectrometer is about 3 mm. Thus, the spectra give data averaged over tens of fluorinated graphite particles. Such measurements cannot detect possible inhomogeneity in nm size in the sample. The size of the beam spot at the beamline was around 150 µm×100 µm that corresponded to single crystallite/particle. We have described the required experimental details as: “The typical size of graphite crystallites was 0.4 × 0.3 × 0.02 mm.” (part 2.1. Materials) and “The spot size of the photon beam was about 3 mm.” (part 2.2 Measurements).
Details of the XPS spectra analysis were added in the part 2.2. Measurements: “The Casa XPS 2.3.15 software (Casa Software Ltd., Teignmouth, UK) was used for data processing. The C 1s, F 1s, and N 1s spectra were fitted by a product of Gaussian-Lorentzian (7:3) peaks after subtraction of a Shirley background.”
The irradiation experiments were carried out for different areas of the samples, and they showed the same trend in the change in the spectral data depending on the exposure time. This information was introduced in the part 2.2. Measurements: “The irradiation experiments were repeated at different spots of the samples, and they showed the same trend in the spectral modifications depending on the exposure time.”
- The description of the XPS results in Figure 1 is qualitative. Since the peaks have been fit to multiple components, can the fraction of each component be provided, perhaps in the SI?
Reply: According to the Reviewer request we added the values of binding energies and relative areas of the components obtained from the fitting of the C 1s spectra in Table S1. This was indicated in the text as: “The binding energies and the relative areas of the components are listed in Table S1.”
- Figure 6. The quality of panels a and b is quite poor, particularly compared to the other figures. Please improve.
Reply: We thank the Reviewer for the careful inspection of the manuscript. We improved the quality of panels a and b in Figure 6 accordingly.
Typos, English
Line 33, can be replaced by
Line 60, This work is aimed at an in situ…
Line 62, samples differing by…
Line 214, assigned to the electron…
Line 274, the remaining fluorine atoms
Line 286, the released nitrogen..
Line 305, the molecules are not detected after …
Line 416, caused the preferred breakage of the latter bonds…
Reply: We thank the Reviewer for the correction of language. The necessary correction were made in the manuscript and they are highlighted in blue.